# CE-LoRA: Computation-Efficient LoRA Fine-Tuning for Large Language Models

## Abstract

Large Language Models (LLMs) demonstrate exceptional performance across various tasks but demand substantial computational resources even for fine-tuning computation. Although Low-Rank Adaptation (LoRA) significantly alleviates memory consumption during fine-tuning, its impact on computational cost reduction is limited. This paper identifies the computation of activation gradients as the primary bottleneck in LoRA's backward propagation and introduces the **C**omputation-**E**fficient **LoRA** (**CE-LoRA**) algorithm, which enhances computational efficiency while preserving memory efficiency. CE-LoRA leverages two key techniques: Approximated Matrix Multiplication, which replaces dense multiplications of large and complete matrices with sparse multiplications involving only critical rows and columns, and the Double-LoRA technique, which reduces error propagation in activation gradients. Theoretically, CE-LoRA converges at the same rate as LoRA, $\mathcal{O}(1/\sqrt{T})$, where $T$ is the number of iterations. Empirical evaluations confirm that CE-LoRA significantly reduces computational costs compared to LoRA without notable performance degradation.

## 1 Introduction

Large Language Models (LLMs) have garnered significant attention in recent years for their exceptional performance across a wide range of practical tasks, including machine translation, commonsense reasoning, and planning, among others Bommasani et al. (2021). The versatility of these models has also driven a growing demand for fine-tuning them on specific tasks or domains to unlock their full potential Zhang et al. (2023); Han et al. (2024b). However, fine-tuning these models remains a highly resource-intensive process, demanding substantial computational power and large amounts of GPU memory. As model parameters and training tokens scale up, the increasing training costs have made it difficult for most organizations to keep pace with advancements in LLM research due to resource constraints.

To tackle these challenges, recent advancements such as Low-Rank Adaptation (LoRA) (Hu et al., 2021) have demonstrated promising results in reducing memory consumption during LLM fine-tuning, enabling the fine-tuning of models with more parameters or larger batch sizes within constrained resources. While LoRA significantly alleviates memory requirements, its reduction of computational costs remains limited — although the low-rank adapters can save part of the computation costs by reducing the matrix sizes, the size of original weight matrices used to calculate *activation gradients* remains unchanged, which contributes to half of the total computation cost in the backpropagation of the original model. The formulation of this computation is illustrated in Section 2. Limited by this computational bottleneck, LoRA can reduce computation by at most half during the backpropagation process. This limitation raises the following open question:

> *Compared with vanilla LoRA, can we develop a more computation-efficient fine-tune algorithm by the same memory budget without sacrificing the statistical efficiency (i.e., convergence)?*

To answer this question, we first conduct a computational analysis of LoRA's backward propagation procedure and identify the primary computational bottleneck as the *calculation of the activation gradients*. This step accounts for the majority of the backward computation load, especially when LoRA employs a relatively small rank $r$. Based on this analysis, we propose the Approximated Matrix Multiplication (AMM) technique

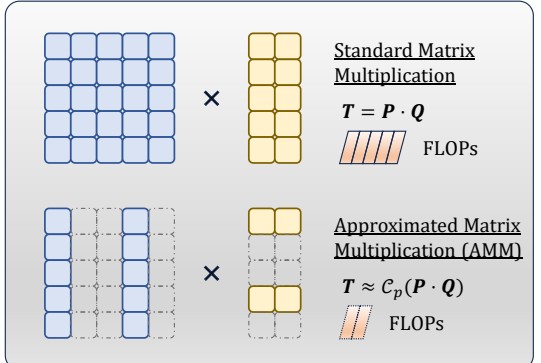 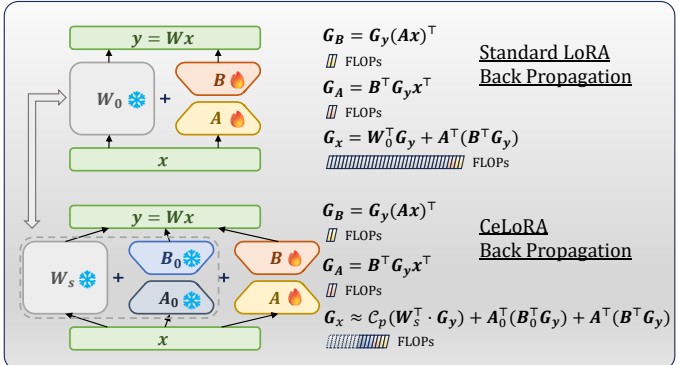

Figure 1: An illustration of the Approximated Matrix Multiplication (AMM) (left) and the CE-LoRA (right).

to reduce the computation of activation gradients. In order to reduce the computation of a dense matrix multiplication $\mathbf{T} = \mathbf{PQ}$, where $\mathbf{P} \in \mathbb{R}^{m \times n}$ and $\mathbf{Q} \in \mathbb{R}^{n \times k}$, AMM directly reduces the size of the matrices by discarding unimportant rows or columns, obtaining $\mathbf{P}[:, \mathcal{I}]$ and $\mathbf{Q}[\mathcal{I}, :]$, as illustrated in Figure 1 (left). Compared to the original multiplication, AMM reduces the arithmetic cost (FLOPs) to $|\mathcal{I}|/n$ of the original by multiplying $\mathbf{P}[:, \mathcal{I}]$ with $\mathbf{Q}[\mathcal{I}, :]$, trading computational accuracy for computational efficiency. To identify the important rows or columns, we compute the importance scores $\alpha_i = \|\mathbf{P}[:, i]\mathbf{Q}[i, :]\|_F$ every $\tau$ iterations and select the top ones in the following $\tau$ times of AMM operation.

Unfortunately, the computational error in the activations' gradients can further propagate to previous layers. Under such a long-lasting effect of inaccurate activation gradients, the accumulated computational error can lead to poor gradient estimations, which severely harms the optimization procedure and degrades the final model performance. To resolve this issue, we develop a double-LoRA technique that significantly reduces the relative error induced to the activation gradients by AMM. Specifically, double-LoRA splits the frozen dense weight matrix into two parts, where the first part applies AMM to save computation, the second part is a frozen LoRA adapter which is computation-efficient without using AMM. In order to reduce the AMM-induced error, we expect the first part to include as little information as possible. Consequently, we initially conduct SVD of the parameter matrix and identify the first part according to the smallest singular values and corresponding vectors.

We evaluate the proposed CE-LoRA algorithm both theoretically and empirically. In theory, we prove that CE-LoRA with momentum SGD converges at a rate of $\mathcal{O}(1/\sqrt{T})$, the same order of LoRA's convergence rate. Empirically, we validate that CE-LoRA can converge at a comparable precision as standard LoRA, with slightly reduced memory consumption and a $3.39\times$ acceleration in computation. To our knowledge, CE-LoRA is the *first* algorithm that accelerates LoRA without sacrificing memory-efficiency or leading to notable performance degradation.

The main contributions of this paper are as follows:

**Contribution 1.** We propose CE-LoRA, a computation-efficient LoRA variant that accelerates fine-tuning by reducing LoRA's dominant backward matrix multiplication. Our design has two key components: (**i**) an AMM operator that approximates this bottleneck multiplication by selecting a subset of indices along the inner dimension; and (**ii**) a double-LoRA error-control mechanism that decomposes the frozen weights into (a) a low-rank adapter part and (b) the remaining residual part, applying AMM only to the latter. Overall, CE-LoRA delivers substantial backward speedups while preserving the memory efficiency of standard LoRA.

**Contribution 2.** We provide a convergence analysis under the standard stochastic optimization assumptions, together with an additional CE-LoRA-specific assumption, which is validated empirically. Concretely, we theoretically prove that CE-LoRA converges with rate $\mathcal{O}(1/\sqrt{T})$, which is the same order as standard LoRA.

**Contribution 3.** We then conduct extensive experiments on various LLMs (e.g., including LLaMA-2-13B and LLaMA-3.1-8B). We experimentally validate that CE-LoRA has a $3.39\times$ acceleration compared with standard LoRA without sacrificing memory efficiency or leading to notable training performance degradation.

## 2    Preliminaries

### 2.1    LoRA Algorithm

In order to fine-tune language models memory-efficiently, LoRA applies a low-rank adapter to each linear layer in the model. Specifically, let $\mathbf{y} = \mathbf{W}\mathbf{x}$ represent a linear layer with $\mathbf{y} \in \mathbb{R}^{m \times b}$, $\mathbf{W} \in \mathbb{R}^{m \times n}$ and $\mathbf{x} \in \mathbb{R}^{n \times b}$, where $m, n$ represent the output and input dimension, respectively, and $b$ represents the batch-size. The LoRA adapter is given by $\mathbf{W} = \mathbf{W}_0 + \mathbf{B}\mathbf{A}$ with $\mathbf{W}_0 \in \mathbb{R}^{m \times n}$ fixed as the pre-trained weights, $\mathbf{B} \in \mathbb{R}^{m \times r}$ and $\mathbf{A} \in \mathbb{R}^{r \times n}$ trainable.

$$\mathbf{y} = \mathbf{W}\mathbf{x}, \tag{Original}$$
$$\mathbf{y} = (\mathbf{W}_0 + \mathbf{B}\mathbf{A})\mathbf{x}. \tag{LoRA}$$

When $r \ll \min\{m, n\}$, the number of trainable parameters in LoRA, $(m + n)r$, is much fewer than that in full fine-tuning, *i.e.*, $mn$, which significantly reduces the memory consumption of the optimizer states.

### 2.2    Computational Bottleneck

While applying LoRA can significantly reduce the memory cost of fine-tuning large language models by decreasing the number of trainable parameters, its backpropagation computation is not reduced proportionally. We analyze the backward pass of the linear layer defined in Section 2. Let $\mathbf{G}_{\boldsymbol{\theta}}$ denote the stochastic gradient of $\boldsymbol{\theta}$ computed by backpropagation (e.g., $\mathbf{G}_{\mathbf{y}} = \partial \mathcal{L} / \partial \mathbf{y}$). Given the upstream gradient $\mathbf{G}_{\mathbf{y}}$, backpropagation computes gradients of the trainable parameters for parameter updates, and the input gradient $\mathbf{G}_{\mathbf{x}}$ for gradient propagation to the previous layer. For the original linear layer $\mathbf{y} = \mathbf{W}\mathbf{x}$, we compute:

$$\mathbf{G}_{\mathbf{W}} = \mathbf{G}_{\mathbf{y}}\mathbf{x}^{\top} \tag{1} \qquad\qquad \mathbf{G}_{\mathbf{x}} = \mathbf{W}^{\top}\mathbf{G}_{\mathbf{y}} \tag{2}$$

where both equation 1 and equation 2 are dense matrix multiplications requiring $2bmn$ FLOPs each, *i.e.*, $4bmn$ FLOPs in total. Accordingly, for LoRA with $\mathbf{W} = \mathbf{W}_0 + \mathbf{B}\mathbf{A}$ and $\mathbf{z} = \mathbf{A}\mathbf{x} \in \mathbb{R}^{r \times b}$, we compute:

$$\mathbf{G}_{\mathbf{B}} = \mathbf{G}_{\mathbf{y}}\mathbf{z}^{\top} \tag{3} \qquad\qquad \mathbf{G}_{\mathbf{z}} = \mathbf{B}^{\top}\mathbf{G}_{\mathbf{y}} \tag{4}$$

$$\mathbf{G}_{\mathbf{A}} = \mathbf{G}_{\mathbf{z}}\mathbf{x}^{\top} \tag{5} \qquad\qquad \mathbf{G}_{\mathbf{x},1} = \mathbf{A}^{\top}\mathbf{G}_{\mathbf{z}} \tag{6}$$

$$\mathbf{G}_{\mathbf{x},2} = \mathbf{W}_0^{\top}\mathbf{G}_{\mathbf{y}} \tag{7} \qquad\qquad \mathbf{G}_{\mathbf{x}} = \mathbf{G}_{\mathbf{x},1} + \mathbf{G}_{\mathbf{x},2} \tag{8}$$

Here equation 3–equation 7 incur $2brm$, $2brm$, $2brn$, $2brn$, and $2bmn$ FLOPs, respectively, while equation 8 is an elementwise addition costing $2bn$ FLOPs. Thus, the total cost is $4br(m + n) + 2bmn + 2bns$. When $r \ll \min\{m, n\}$, the LoRA-specific terms are lower-order, and the backward computation is still dominated by the dense multiplication $\mathbf{W}_0^{\top}\mathbf{G}_{\mathbf{y}}$ in equation 7, which alone requires $2bmn$ FLOPs. As a result, LoRA reduces the backward FLOPs by only about half compared to full fine-tuning.

## 3    CE-LoRA: Computation-Efficient LoRA

### 3.1    Approximated Matrix Multiplication (AMM)

Consider matrix multiplication $\mathbf{T} = \mathbf{P}\mathbf{Q}$, where $\mathbf{T} \in \mathbb{R}^{m \times k}$, $\mathbf{P} \in \mathbb{R}^{m \times n}$ and $\mathbf{Q} \in \mathbb{R}^{n \times k}$. Let $\mathbf{p}_1, \mathbf{p}_2, \cdots, \mathbf{p}_n$ denote the column vectors of matrix $\mathbf{P}$, and $\mathbf{q}_1, \mathbf{q}_2, \cdots, \mathbf{q}_n$ denote the column vectors of matrix $\mathbf{Q}^{\top}$. We can rewrite the matrix multiplication into:

$$\mathbf{T} = \sum_{i=1}^{n} \mathbf{p}_i \mathbf{q}_i^{\top}.$$

To estimate the product $\mathbf{T}$ computation-efficiently, we may assume the matrices $\mathbf{P}$ and $\mathbf{Q}$ enjoy some kinds of structured sparsity, such that a few $(\mathbf{p}_i\mathbf{q}_i^{\top})'s$ contribute to most of the result $\sum_{i=1}^{n} \mathbf{p}_i\mathbf{q}_i^{\top}$, in which case we could estimate $\mathbf{T}$ by computing the most important parts only. Specifically, we identify $s$ most important indices $1 \le i_1 < \cdots < i_s \le n$, and the AMM estimate of $\mathbf{T}$ is given by:

$$\hat{\mathbf{T}} = \sum_{j=1}^{s} \mathbf{p}_{i_j} \mathbf{q}_{i_j}^{\top} = \hat{\mathbf{P}}\hat{\mathbf{Q}},$$

where $\hat{\mathbf{P}}$ and $\hat{\mathbf{Q}}^{\top}$ collect column vectors $\{\mathbf{p}_{i_j}\}_{j=1}^{s}$ and $\{\mathbf{q}_{i_j}\}_{j=1}^{s}$, respectively. The efficiency of AMM is concerned with the number of selected indices $s$, or the structured sparsity $p := s/n \in (0, 1]$. Replacing the

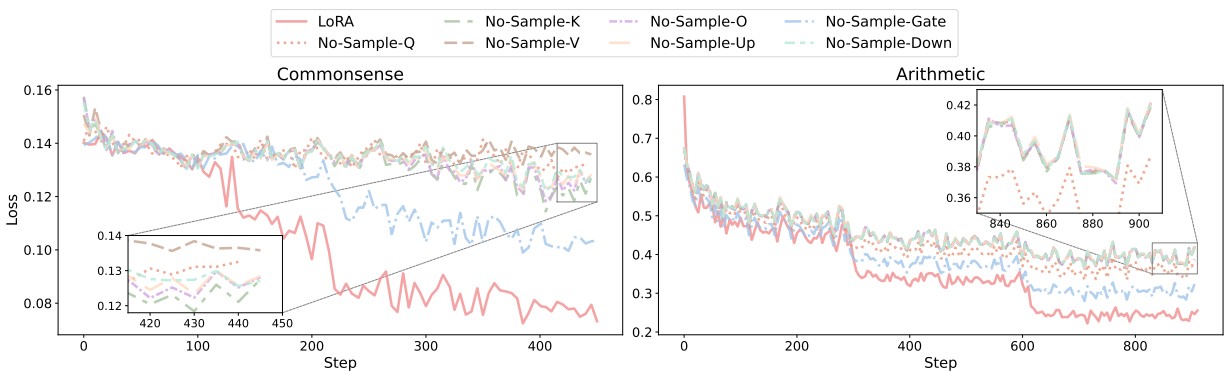

Figure 2: Layer-wise sensitivity analysis of LLaMA3.2-1B.

dense matrix multiplication $\mathbf{T} = \mathbf{PQ}$ by AMM estimate $\hat{\mathbf{T}} = \hat{\mathbf{P}}\hat{\mathbf{Q}}$, the computational complexity is reduced from $2mnk$ to $2msk = p \cdot (2mnk)$. Hereafter, we use $\mathcal{C}_p(\mathbf{P} \cdot \mathbf{Q})$ to denote the AMM estimate of matrix multiplication $\mathbf{P} \cdot \mathbf{Q}$ with structured sparsity $p$.

An important question is how to select the indices $\mathcal{I} = \{i_1, i_2, \cdots, i_s\}$ properly. A previous research (Drineas et al., 2006) has studied a random sampling strategy, which does not work well in our experiments. Based on the above intuition, we define the importance score $\alpha_i$ of index $i$ by the Frobenius norm $\|\mathbf{p}_i \mathbf{q}_i^\top\|_F$ and attempt to select the indices with highest scores. However, as calculating $\{\alpha_i\}_{i=1}^k$ requires the same amount of computation as that of conducting the original matrix multiplication, we cannot determine $\mathcal{I}$ based on the calculation results of $\{\alpha_i\}_{i=1}^k$ in every iteration. We use historical information to mitigate this issue. Specifically, the matrices $\mathbf{P}, \mathbf{Q}$ we multiply by AMM should be variables that live along the whole optimization process, and $\mathbf{P}^t, \mathbf{Q}^t$ are multiplied at every iteration $t$. The corresponding $\mathcal{I}^t$ is only re-selected according to the top-$s$ importance scores every $\tau$ iterations and is reused in intermediate ones. To reduce the computational bottleneck in LoRA's backward propagation, we apply AMM to step equation 7:

$$\mathbf{G}_{\mathbf{x},2} = \mathcal{C}_p(\mathbf{W}_0^\top \cdot \mathbf{G}_{\mathbf{y}}). \tag{9}$$

## 3.2 Double-LoRA Mechanism

Although computation-efficient, AMM will induce errors to $g_x$, the gradient with respect to the activations. These errors propagate backward through the network, potentially compounding as they traverse previous layers. If the magnitude of these errors is not properly controlled, the accuracy of the parameter gradients can be significantly degraded. To mitigate this issue, we propose a double-LoRA mechanism to alleviate the error induced by the AMM operation in each layer. Intuitively, we wish the objective matrix multiplication result we estimate by AMM has as little contribution to the activation gradient as possible. This drives us to further separate the frozen matrix $\mathbf{W}_0$ into two parts: a low-rank part inheriting computational efficiency without AMM, and a residual part with a relatively small magnitude.

Specifically, we initially compute the SVD of $\mathbf{W}_0$, yielding:

$$\mathbf{W}_0 = \mathbf{U}\boldsymbol{\Sigma}\mathbf{V}^\top$$

Next, We collect the principal low-rank component $\mathbf{B}_0 = \mathbf{U}[:r]\boldsymbol{\Sigma}^{1/2}$, $\mathbf{A}_0 = \boldsymbol{\Sigma}^{1/2}(\mathbf{V}[:r])^\top$, and the residual $\mathbf{W}_s = \mathbf{W}_0 - \mathbf{B}_0\mathbf{A}_0$. By separating $\mathbf{W}_0$ to $\mathbf{W}_s + \mathbf{B}_0\mathbf{A}_0$, we split the matrix to two parts. The first part $\mathbf{W}_s$ is believed to have better structured sparsity and is more compatible to AMM. The second low-rank part $\mathbf{B}_0\mathbf{A}_0$ is computation-efficient just like the trainable LoRA adapter $\mathbf{BA}$. Combining AMM with double-LoRA, equation 9 is further replaced by:

$$\mathbf{G}_{\mathbf{x},2} = \mathcal{C}_p(\mathbf{W}_s^\top \cdot \mathbf{G}_{\mathbf{y}}) + \mathbf{A}_0^\top(\mathbf{B}_0^\top \mathbf{G}_{\mathbf{y}}). \tag{10}$$

## 3.3 Layer-wise Adaptive Sparsity

It is natural to apply more aggressive sparsity to layers that are relatively robust to computational errors, while using more conservative sparsity for those that are more sensitive.

---

**Algorithm 1** CE-LoRA

---

**Input:** Frozen layer weight $\mathbf{W}_\ell \in \mathbb{R}^{m_\ell \times n_\ell}$, sparsity level $p_\ell$, double-LoRA rank $r_{0,\ell}$, indices recomputing period $\tau$, Top-K indices $\mathcal{I}_\ell =$ empty, optimizer $\rho$.

---

**Initialize Double-LoRA**
1:    **for** Layer $\ell = 1, 2, \cdots, L$ **do**
2:      Conducting SVD on frozen weight matrix
     $\mathbf{W}_{0,\ell} = \mathbf{U}_\ell \boldsymbol{\Sigma}_\ell \mathbf{V}_\ell^\top$;
3:      $\mathbf{A}_{0,\ell}, \mathbf{B}_{0,\ell} \leftarrow \sqrt{\boldsymbol{\Sigma}_\ell}\mathbf{V}_{\ell\,[:r_0,]}^\top, \mathbf{U}_{\ell[:,:r_0]}\sqrt{\boldsymbol{\Sigma}_\ell}$ ;            ▷ Stored in layer's buffer.
4:    **end for**

---

5: **for** CE-LoRA Training Step $t = 0, 1, \cdots, T-1$ **do**
6:    **for** Layer $\ell = 1, 2, \cdots, L$ **do**                                    ▷ **Forward**
7:      $\mathbf{z}_\ell \leftarrow \mathbf{A}_\ell \mathbf{x}_\ell$;
8:      $\mathbf{y}_\ell \leftarrow \mathbf{W}_{0,\ell}\mathbf{x}_\ell + \mathbf{B}_\ell \mathbf{z}_\ell$;
9:    **end for**
10:    **for** Layer $\ell = L, L-1, \cdots, 1$ **do**                            ▷ **Backward**
11:      $\mathbf{W}_{s,\ell} \leftarrow \mathbf{W}_{0,\ell} - \mathbf{B}_{0,\ell}\mathbf{A}_{0,\ell}$;
12:      **if** $\tau \mid t$ **or** $\mathcal{I}_\ell$ is empty **then**
13:        $\alpha_{i,\ell} \leftarrow \left\| \mathbf{W}_{s,\ell[:,i]}^\top \mathbf{G}_{\mathbf{y}_\ell[i,:]} \right\|_F, \ \forall i \in \{1, \ldots, m_\ell\}$
14:      Select $\left\{ i_{1,\ell}, \cdots, i_{\mathrm{K}_\ell,\ell} \right\}$ with largest $\alpha_{i,\ell}$'s;
15:      $\mathcal{I}_\ell = \left\{ i_{1,\ell}, \cdots, i_{\mathrm{K}_\ell,\ell} \right\}$;                           ▷ Here $\mathrm{K}_\ell = \lceil m_\ell p_\ell \rceil$
16:      **end if**
17:      $\mathbf{G}_{\mathbf{B}_\ell} \leftarrow \mathbf{G}_{\mathbf{y}_\ell}\mathbf{z}_\ell^\top$;
18:      $\mathbf{G}_{\mathbf{z}_\ell} \leftarrow \mathbf{B}_\ell^\top \mathbf{G}_{\mathbf{y}_\ell}$;
19:      $\mathbf{G}_{\mathbf{A}_\ell} \leftarrow \mathbf{G}_{\mathbf{z}_\ell}\mathbf{x}_\ell^\top$;
20:      $\mathbf{G}_{\mathbf{x}_\ell} \leftarrow \mathbf{W}_{s,\ell[,\mathcal{I}]}^\top \mathbf{G}_{\mathbf{y}_\ell[\mathcal{I},]} + \mathbf{A}_{0,\ell}^\top(\mathbf{B}_{0,\ell}^\top \mathbf{G}_{\mathbf{y}_\ell}) + \mathbf{A}_\ell^\top \mathbf{G}_{\mathbf{z}_\ell}$;
21:    **end for**
22: Use optimizer $\rho$ to update $\{\mathbf{A}_\ell, \mathbf{B}_\ell\}_{\ell=1}^L$ according to $\{\mathbf{G}_{\mathbf{A}_\ell}, \mathbf{G}_{\mathbf{B}_\ell}\}_{\ell=1}^L$;
23: **end for**

---

Inspired by Hu et al. (2025); Ma et al. (2024); Jaiswal et al. (2024); Zeng et al. (2024); Malinovskii et al. (2024); Zhang et al. (2024); Liu et al. (2024), we adopt a layer-wise adaptive sparsity strategy for CE-LoRA.

To determine which layers are more sensitive to varying sparsity levels, we conduct experiments on two small fine-tuning datasets: the Commonsense 14K dataset and the Math 7K dataset Hu et al. (2023). In these experiments, we fix LoRA's rank to 32, and set both CE-LoRA's trainable LoRA rank and its frozen Double-LoRA rank to 28. For each CE-LoRA configuration, we vary the sparsity level of one layer type, while setting the sparsity of all remaining layer types to $p = 0.3$. As shown in Figure 2, the `Gate` layers are essential for preventing error propagation. In addition, the `Q` and `K` layers have a strong impact on arithmetic and commonsense reasoning tasks, respectively. Based on these findings, we disable sparsity for the `Q`, `K`, and `Gate` layers. For the remaining MHA layers, we use $p = 0.55$, and for the last two layers in the FFN, we set $p = 0.65$ throughout our experiments.

Table 1: Computation and memory analysis for a single linear layer.

| Method | Standard AdamW | LoRA | CE-LoRA |
|---|---|---|---|
| **Memory Usage** | $10mn + 2bm$ | $2mn + 10r(m+n)$ $+ 2b(m+r)$ | $2mn + 2r_0(m+n)$ $+ 10r(m+n) + 2b(m+r)$ |
| **Backward Computation** | $4bmn$ | $2bmn + 4br(m+n)$ | $(2pb+1)mn$ $+ 2(r_0 + br_0 + 2br)(m+n)$ |
| **Forward Computation** | $2bmn$ | $2bmn + 2br(m+n)$ | $2bmn + 2br(m+n)$ |

### 3.4 Algorithm

Overall, CE-LoRA integrates AMM, double-LoRA, and layer-wise adaptivity, as outlined in Algorithm 1. During model initialization, we replace all frozen linear layers with CE-LoRA and apply the double-LoRA technique to the weight matrix $\mathbf{W}_0$, resulting in low-rank components $\mathbf{A}_0$ and $\mathbf{B}_0$ (lines 2–3). For each training step $t$, the forward pass of a CE-LoRA linear layer behaves the same as the original frozen linear layer (lines 8). In the backward pass, CE-LoRA first computes the residual weight matrix by subtracting the low-rank components from the original weight matrix (line 11). Next, if the current step $t$ is a multiple of $\tau$ or if the indices are empty (*e.g.*, at the start of training), the top-K indices are updated (lines 12–15). Finally, CE-LoRA uses AMM to compute activation gradient $\mathbf{G}_{\mathbf{x}_\ell}$ (line 20).

### 3.5 Complexity Analysis

To better illustrate the computational efficiency of CE-LoRA, we theoretically compare the computational and memory complexity of CE-LoRA with LoRA and standard AdamW fine-tuning. Consider linear layer $\mathbf{y} = \mathbf{W}\mathbf{x}$ with $\mathbf{W} \in \mathbb{R}^{m \times n}$, trained with LoRA rank $r$, double-LoRA rank $r_0$, structured sparsity $p$ and batch size $b$ using BF16 precision. As illustrated in Table 1, CE-LoRA can achieve a memory usage similar to LoRA by applying slightly smaller $r_0$ and $r$, while significantly reduce the backward computation by applying a relatively small $p$ when $b \gg 1$ and $r \ll \min\{m, n\}$. When combined with low-precision training, the influence of double-LoRA can be further reduced, as the frozen low-rank parameters do not require high-precision weight copies or gradient accumulators.

## 4 Convergence Analysis

We first present the assumptions under which we prove CE-LoRA's convergence properties.

**Assumption 4.1** (Lower Boundedness). Loss function $f : \mathbb{R}^d \to \mathbb{R}$ satisfies $\inf_{\mathbf{x} \in \mathbb{R}^d} f(\mathbf{x}) > -\infty$.

**Assumption 4.2** (L-Smoothness). Loss function $f$ is $L$-smooth, *i.e.*, it holds $\forall \mathbf{x}, \mathbf{y} \in \mathbb{R}^d$ that

$$\|\nabla f(\mathbf{x}) - \nabla f(\mathbf{y})\|_2 \le L\|\mathbf{x} - \mathbf{y}\|_2.$$

**Assumption 4.3** (Stochastic Gradient). We assume the stochastic gradient oracle satisfies

$$\mathbb{E}[\nabla F(\mathbf{x}^t; \xi^t)] = \nabla f(\mathbf{x}^t); \tag{11}$$

$$\mathbb{E}[\|\nabla F(\mathbf{x}^t; \xi^t) - \nabla f(\mathbf{x}^t)\|^2] \le \sigma^2, \tag{12}$$

for some $\sigma > 0$.

Assumptions 4.1-4.3 are standard assumptions commonly used in stochastic optimization.

**Assumption 4.4** (Gradient Error). Let $\mathbf{g}^t$ and $\hat{\mathbf{g}}^t$ denote the original stochastic gradient $\nabla F(\mathbf{x}^t, \xi^t)$ and its estimation by CE-LoRA, it holds that

$$\|\hat{\mathbf{g}}^t - \mathbf{g}^t\|^2 \le (1 - \delta)\|\mathbf{g}^t\|^2, \tag{13}$$

and

$$\|\mathbb{E}_{\xi^t \sim \mathcal{D}}[\hat{\mathbf{g}}^t] - \nabla f(\mathbf{x}^t)\|^2 \le (1 - \delta)\|\nabla f(\mathbf{x}^t)\|^2, \tag{14}$$

for some $\delta \in (0, 1]$.

Assumption 4.4 illustrates the property of stochastic gradients in CE-LoRA. Though not standard, this assumption can be empirically justified by our experimental results.

**Empirical Justification of Assumption 4.4.** To justify equation 13, we conduct experiments on language model fine-tuning tasks on google/gemma-2b Team et al. (2024) model using CoLA Warstadt et al. (2019), RTE and MRPC Dolan & Brockett (2005) datasets, three tasks in the GLUE benchmark Wang et al. (2019). In these experiments, we use AdamW with a learning rate of `1e-5` to train for 1 epoch per task and calculate the relative error $\|\hat{\mathbf{g}}^t - \mathbf{g}^t\|^2 / \|\mathbf{g}^t\|^2$ every 10 steps, as illustrated in Figure 3, where all relative errors are below

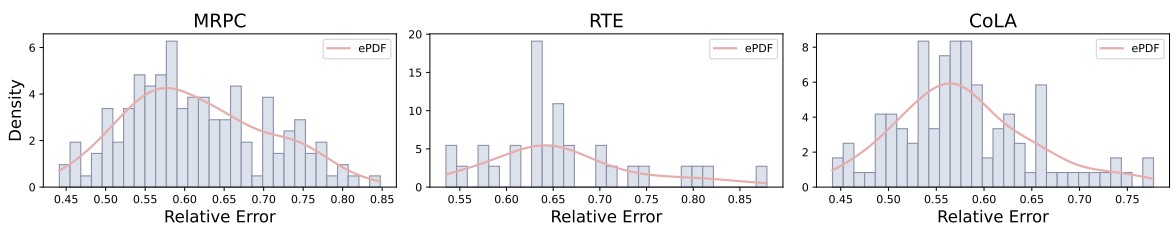

Figure 3: Empirical validation of equation 13 on MRPC (left), RTE (middle) and CoLA (right).

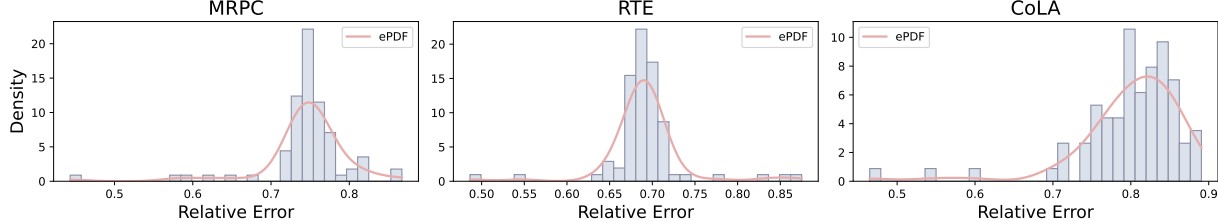

Figure 4: Empirical validation of equation 14 on MRPC (left), RTE (middle) and CoLA (right).

0.9. To justify equation 14, we conduct experiments on the same model and datasets, where we alternatively calculate one iteration of full gradient AdamW and one epoch of random gradient AdamW, each with a learning rate of `1e-5` for a total of 80 cycles. We apply a rank of 64 for both LoRA and double-LoRA in CE-LoRA, and apply a structured sparsity of $p_{\text{FFN}} = 0.9$ and $p_{\text{MHA}} = 0.4$. We calculate the relative error $\|\mathbb{E}_{\xi^t \sim \mathcal{D}}[\hat{\mathbf{g}}^t] - \nabla f(\mathbf{x}^t)\|^2 / \|\nabla f(\mathbf{x}^t)\|^2$ for every full-gradient step, as illustrated in Figure 4, where all relative errors are below 0.9.

We now propose the convergence results of CE-LoRA using the momentum SGD optimizer with the following update:

$$\mathbf{m}^t = (1 - \beta_1)\mathbf{m}^{t-1} + \beta_1 \hat{\mathbf{g}}^t,$$
$$\mathbf{x}^{t+1} = \mathbf{x}^t - \eta \mathbf{m}^t,$$

**Theorem 4.5.** *Under Assumptions 4.1 - 4.4, if $\beta_1 \in \left(0, \frac{\delta}{24-12\delta}\right)$ and $\eta \leq \min\left\{\frac{L}{2}, \frac{\beta_1}{L} \cdot \sqrt{\frac{\delta}{8}}\right\}$, CE-LoRA with momentum SGD converges as*

$$\frac{1}{T+1}\sum_{t=0}^{T} \mathbb{E}[\|\nabla f(\mathbf{x}^t)\|_2^2] \leq \frac{8[f(\mathbf{x}^0) - \inf_{\mathbf{x}} f(\mathbf{x})]}{\delta \eta (T+1)} + \frac{8\|\mathbf{m}^0 - \nabla f(\mathbf{x}^0)\|_2^2}{\delta \beta_1 (T+1)} + \frac{24\beta_1 \sigma^2}{\delta}.$$

**Corollary 4.6.** *Under Assumptions 4.1-4.4, if we choose $\beta_1 = \left(\frac{24}{\delta} + \sigma\sqrt{\frac{\delta^{1/2}(T+1)}{L\Delta}}\right)^{-1}$, $\eta = \left(2L + \frac{2^{3/2}L}{\delta^{1/2}\beta_1}\right)^{-1}$, CE-LoRA with momentum SGD converges as*

$$\frac{1}{T+1}\sum_{t=0}^{T} \mathbb{E}[\|\nabla f(\mathbf{x}^t)\|_2^2] = \mathcal{O}\left(\frac{L\Delta}{\delta^{5/2}(T+1)} + \sqrt{\frac{L\Delta\sigma^2}{\delta^{5/2}(T+1)}}\right),$$

*where $\Delta := f(\mathbf{x}^0) - \inf_{\mathbf{x}} f(\mathbf{x}) + (\delta/L) \cdot \|\mathbf{m}^0 - \nabla f(\mathbf{x}^0)\|_2^2$.*

Detailed proofs are deferred to Appendix A.

## 5 Experiments

In this section, we present a comprehensive set of experiments to evaluate the convergence performance and computational efficiency of CE-LoRA, and compare it against the baseline method.

### 5.1 Experimental Setup

**Datasets.** We follow the benchmark design outlined in Hu et al. (2023) and evaluate CE-LoRA on two popular reasoning benchmarks:

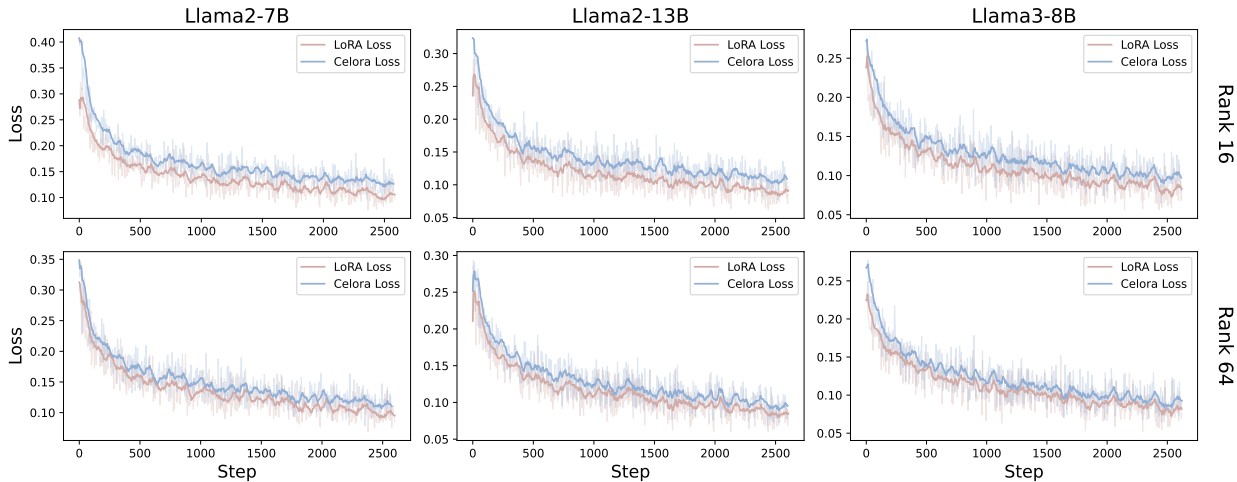

Figure 5: Loss curve of commonsense reasoning fine-tuning. Each row corresponds to a trainable parameter setting, while each column represents base models: LLaMA2-7B/13B and LLaMA3.1-8B.

- **Commonsense Reasoning**: This dataset includes eight tasks: BoolQ Clark et al. (2019), PIQA Bisk et al. (2020), SocialQA Sap et al. (2019), HellaSwag Zellers et al. (2019), WinoGrande Sakaguchi et al. (2021), ARC-challenge Clark et al. (2018), ARC-easy Clark et al. (2018), and OpenbookQA Mihaylov et al. (2018). In our experiments, we fine-tune all models using the Commonsense 170K dataset Hu et al. (2023), which is constructed by combining the training sets from these eight tasks.

- **Arithmetic Reasoning**: This benchmark consists of seven subsets: MultiArith Roy & Roth (2016), GSM8K Cobbe et al. (2021), AddSub Hosseini et al. (2014), AQuA Ling et al. (2017), SingleEq Koncel-Kedziorski et al. (2015), SVAMP Patel et al. (2021) and MAWPS Koncel-Kedziorski et al. (2016). We fine-tune the models on the Math 10k dataset Hu et al. (2023), which includes training data from GSM8K, MAWPS, and AQuA, augmented by language models with chain-of-thought reasoning steps.

**Fine-tuned models and hyper-parameters.** We fine-tune LLaMA-2-7B, LLaMA-2-13B Touvron et al. (2023b), and LLaMA-3.1-8B Grattafiori et al. (2024) using both CE-LoRA and LoRA. The adapter is applied to all linear layers in each transformer block, including Q, K, V, O, Up, Gate, and Down. Unless specified otherwise, all CE-LoRA experiments replace the frozen V, O, Up, and Down layers with CE-LoRA layers. The sparsity levels are set as follows: $p_V = p_O = 0.55$ and $p_{Up} = p_{Down} = 0.65$. For consistency, the same set of hyperparameters is applied across both methods for each model size. All experiments are conducted using the BF16 format to optimize memory usage.

## 5.2 Statistical Efficiency of CE-LoRA

In this set of experiments, we evaluate the convergence performance of CE-LoRA using two critical metrics: the accuracy achieved on each benchmark and the trajectory of the fine-tuning loss across training iterations. By monitoring these metrics, we aim to gain insights into how quickly and effectively CE-LoRA converges compared to LoRA.

Table 2: Comparison among eight commonsense reasoning tasks for the LLaMA2-7B/13B, LLaMA3.1-8B

| Model | Method | Rank | BoolQ | PIQA | SIQA | HellaSwag | Wino | ARC-e | ARC-c | OBQA | Avg. ↑ |
|-------|--------|------|-------|------|------|-----------|------|-------|-------|------|--------|
| LLaMA2-7B | LoRA | 16 | 71.99 | 84.49 | 81.73 | 94.45 | 85.95 | 87.63 | 73.21 | 83.80 | 82.91 |
| | CE-LoRA | 14 | 70.24 | 82.59 | 79.27 | 93.17 | 82.72 | 85.56 | 70.65 | 79.80 | 80.50 |
| | LoRA | 64 | 72.26 | 84.88 | 82.70 | 94.97 | 86.42 | 88.55 | 74.74 | 86.40 | 83.87 |
| | CE-LoRA | 56 | 71.68 | 85.20 | 82.09 | 94.61 | 83.98 | 87.29 | 73.29 | 84.20 | 82.79 |
| LLaMA2-13B | LoRA | 16 | 75.32 | 88.03 | 83.21 | 81.14 | 92.34 | 88.20 | 96.08 | 88.71 | 86.63 |
| | CE-LoRA | 14 | 73.21 | 86.62 | 82.14 | 94.65 | 86.27 | 90.15 | 77.22 | 84.80 | 84.38 |
| | LoRA | 64 | 75.72 | 88.85 | 84.39 | 96.34 | 88.71 | 92.42 | 81.83 | 89.60 | 87.23 |
| | CE-LoRA | 56 | 74.01 | 86.51 | 83.11 | 92.74 | 87.92 | 91.37 | 79.61 | 85.40 | 85.08 |
| LLaMA3-8B | LoRA | 16 | 75.84 | 90.86 | 83.52 | 96.93 | 89.90 | 94.07 | 84.47 | 88.8 | 88.05 |
| | CE-LoRA | 14 | 72.08 | 89.72 | 82.65 | 96.24 | 88.32 | 93.35 | 83.36 | 87.60 | 86.66 |
| | LoRA | 64 | 75.63 | 90.21 | 83.32 | 96.38 | 88.95 | 93.39 | 84.04 | 89.20 | 87.64 |
| | CE-LoRA | 56 | 73.36 | 89.66 | 82.40 | 95.76 | 86.42 | 93.14 | 82.68 | 87.60 | 86.38 |

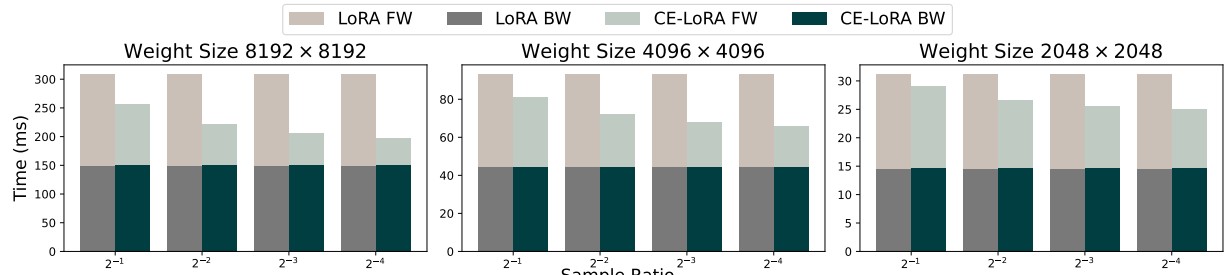

Figure 6: Comparison of training latency for CE-LoRA and LoRA at various sparsity levels (i.e., $\frac{1}{2}$, $\frac{1}{4}$, $\frac{1}{8}$, $\frac{1}{16}$) across three model shapes: $(8192, 8192)$, $(4096, 4096)$, and $(2048, 2048)$. CE-LoRA provides significant speedups in the backward pass, leading to a maximum of 36.3% overall reduction in end-to-end training time compared to LoRA.

**Accuracy**. We trained both CE-LoRA and LoRA under low-rank (LoRA rank of 16, CE-LoRA rank of 14) and high-rank (LoRA rank of 64, CE-LoRA rank of 56) configurations across two reasoning datasets for one epoch. Table 2 and Table 3 summarize the results for the commonsense and arithmetic reasoning benchmarks. The experimental outcomes demonstrate that, across all LoRA rank settings in both benchmarks, CE-LoRA achieves fine-tuning accuracy that is nearly identical to that of LoRA, with an average difference in results of 1.58%. These findings suggest that our approach has a negligible impact on the original LoRA fine-tuning accuracy. The slight differences in accuracy between CE-LoRA and LoRA on the test sets can primarily be attributed to the scaling of CE-LoRA's rank, which was adjusted to ensure a fair experimental comparison.

**Loss curve**. Figure 5 illustrates the loss curves of both CE-LoRA and LoRA under different rank settings across the three models on the commonsense reasoning fine-tuning task. In each setting, CE-LoRA's loss curves nearly overlap with those of its LoRA counterparts, indicating similar convergence behaviors. These results highlight the effectiveness of our method, empirically demonstrating that CE-LoRA can achieve nearly the same convergence capability as the original LoRA while potentially offering computational savings. The overlapping loss curves suggest that CE-LoRA does not introduce additional convergence challenges and maintains training stability comparable to LoRA.

## 5.3 Computation Efficiency

In these experiments, we measure CE-LoRA's training efficiency by comparing the average training step latency of a single-layer CE-LoRA with a single-layer LoRA. All experiments are conducted on a single `NVIDIA-HGX-H20-(96GB)` GPU to maintain consistent hardware conditions. For a fair comparison, both CE-LoRA and LoRA employ the same trainable rank of 64. We run experiments on three different model weight sizes—$(8192, 8192)$, $(4096, 4096)$, and $(2048, 2048)$—using a fixed batch size of 16 and a sequence length of 8192. To measure average training step latency, each configuration is tested over 100 runs. The first 10 iterations of each run are considered warmup and are excluded from latency measurements to mitigate initialization overhead.

Figure 6 compares the results of LoRA and CE-LoRA with various sparsity levels and shows that CE-LoRA achieves a consistent reduction in overall training time, with a maximum of 36.3% speedup. As illustrated, CE-LoRA 's forward pass latency closely matches that of LoRA's due to the unchanged forward logic of the frozen layer. However, in the backward pass, CE-LoRA outperforms LoRA by up to 3.39× with some aggressive sampling rate. The observed improvements in wall-clock speed are primarily attributed to two key factors: (i) CE-LoRA effectively reduces the theoretical floating-point operations required during backpropagation for frozen layers. (ii) We developed specialized CUDA kernels tailored for low-rank computations inherent in CE-LoRA's backpropagation process, which optimize memory access patterns, resulting in enhanced computational efficiency and reduced latency.

Table 3: Performance comparison of LoRA and CE-LoRA on seven arithmetic reasoning tasks.

| Model | Method | Rank | MultiArith | GSM8K | AddSub | AQuA | SingleEq | SVAMP | MAWPS | Avg. ↑ |
|-------|--------|------|-----------|-------|--------|------|----------|-------|-------|--------|
| LLaMA3-8B | LoRA | 16 | 94.50 | 64.59 | 90.89 | 47.24 | 92.13 | 76.30 | 88.66 | 79.19 |
| | CE-LoRA | 14 | 94.00 | 62.09 | 91.14 | 44.88 | 93.50 | 75.00 | 90.76 | 78.77 |
| | LoRA | 64 | 96.33 | 65.50 | 90.63 | 49.61 | 92.91 | 81.2 | 89.50 | 80.81 |
| | CE-LoRA | 56 | 96.17 | 62.02 | 88.86 | 47.64 | 93.31 | 77.10 | 89.08 | 79.17 |

## 6 Related Works

**Large Language Models.** Since the transformer structure was proposed in the famous work Vaswani (2017) in 2017, it has shown great potential in various tasks, including reasoning, planning, machine translation, *etc.*, and has become a popular choice in modern LLM designs, *e.g.*, GPT (Radford, 2018; Radford et al., 2019; Brown et al., 2020), OPT (Zhang et al., 2022), LLaMA (Touvron et al., 2023a;c; Dubey et al., 2024), BLOOM (Le Scao et al., 2023), BERT (Devlin, 2018), Falcon (Penedo et al., 2023), *etc.* In general, the basic structure of a transformer block consists of a multi-head attention (MHA) module followed by a feed-forward network (FFN), combined with normalization and residual connections. Linear layers take up most of the trainable parameters in transformers and account for the expensive training and inference costs.

**Memory-Efficient Training Algorithms.** As the scale of LLM parameters grows, the memory consumption to train these models has become a bottleneck problem. Recent studies have proposed a series of works in order to reduce training-time memory consumption, enabling LLM researchers to effectively pre-train / fine-tune larger LLMs within constrained computational resources. Houlsby et al. (2019); Pfeiffer et al. (2020) fine-tune LLMs parameter-efficiently by adding trainable adapter layers, LoRA (Hu et al., 2021) reparameterizes linear layers in transformers with low-rank adapters, ReLoRA (Lialin et al., 2023) extends to pre-training tasks by accumulating LoRA updates, $S^2$FT (Yang et al., 2024) applies sparse structures, SLTrain (Han et al., 2024a) combines low-rank and sparse structures. Besides the above parameter-efficient approaches, another line of works reduce the memory consumption for optimizer states by optimizing in periodically updated subspaces, including LISA (Pan et al., 2024), GaLore (Zhao et al., 2024), GoLore (He et al., 2024) and FLORA (Hao et al., 2024). In addition, BackRazor (Jiang et al., 2022) and PreBackRazor (Yu et al., 2024) improve memory-efficiency by compressing the activation memory. Furthermore, quantization methods (Micikevicius et al., 2017; Dettmers et al., 2024) that are orthogonal to the above approaches have shown nice compatibilities in memory cost reduction.

**Computation-Efficient Training Algorithms.** Though not specially designed for computational efficiency, a lot of memory-efficient training algorithms, particular those belonging to parameter-efficient fine-tuning (PEFT), can also reduce computational costs to some extent. On the other hand, the training throughput can also be improved by utilizing a larger batch size thanks to the reduced memory consumption (Zhu et al., 2024). However, the computational savings of these approaches are limited by precisely retaining the complete backward propagation process. Recently, Woo et al. (2024) proposes DropBP, an approach orthogonal to PEFT that saves computation by strategically skip connections in backward propagation. Since some layers are dropped during backward propagation, corresponding parameters do not have gradients for update. To the best of our knowledge, this paper provides the *first* approach to accelerate LoRA by employing structured sparsity to reduce the computational bottleneck in backward propagation without sacrificing memory-efficiency or model performance.

## 7 Conclusion

We present CE-LoRA, a computation-efficient variant of LoRA that accelerates fine-tuning by reducing the dominant activation-gradient matrix multiplication in LoRA's backward pass. CE-LoRA integrates (i) an Approximated Matrix Multiplication (AMM) operator that reduces computation by selecting a subset of inner-dimension indices, (ii) a double-LoRA error-control mechanism that separates the computation into a low-rank adapter component and a residual component, applying AMM only to the residual to mitigate approximation-induced gradient errors, and (iii) a layer-wise adaptive sparsity strategy to balance robustness and efficiency across layers. We provide a convergence analysis under standard stochastic optimization assumptions, together with a CE-LoRA-specific assumption, and show that CE-LoRA with momentum SGD achieves an $\mathcal{O}(1/\sqrt{T})$ convergence rate, matching the order of standard LoRA. Empirically, CE-LoRA matches LoRA in fine-tuning accuracy and convergence stability across multiple LLMs and tasks, while delivering substantial efficiency gains—up to a $3.39\times$ acceleration—without introducing notable memory overhead or performance degradation.

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

# A    Proof of Theorem 4.5

In this section, we provide detailed proofs for Theorem 4.5. We first prove the following lemma.

**Lemma A.1.** *Under Assumptions 4.1-4.4, if $\beta_1 \in (0, 1)$, it holds that*

$$\sum_{t=0}^{T} \mathbb{E}[\|\mathbf{m}^t - \nabla f(\mathbf{x}^t)\|_2^2] \leq \frac{2\|\mathbf{m}^0 - \nabla f(\mathbf{x}^0)\|_2^2}{\beta_1} + \frac{4L^2}{\delta\beta_1^2} \sum_{t=1}^{T} \|\mathbf{x}^t - \mathbf{x}^{t-1}\|_2^2$$
$$+ \left(1 - \frac{\delta}{2}\right)(1 + 6\beta_1) \sum_{t=1}^{T} \mathbb{E}[\|\nabla f(\mathbf{x}^t)\|_2^2] + 6T\beta_1\sigma^2. \tag{15}$$

*Proof.* According to the update of momentum, we have

$$\mathbf{m}^t - \nabla f(\mathbf{x}^t) = (1 - \beta_1)(\mathbf{m}^{t-1} - \nabla f(\mathbf{x}^t)) + \beta_1(\hat{\mathbf{g}}^t - \nabla f(\mathbf{x}^t)).$$

Taking expectation we have

$$\mathbb{E}[\|\mathbf{m}^t - \nabla f(\mathbf{x}^t)\|_2^2] = \mathbb{E}[\|(1 - \beta_1)(\mathbf{m}^{t-1} - \nabla f(\mathbf{x}^t)) + \beta_1(\mathbb{E}[\hat{\mathbf{g}}^t] - \nabla f(\mathbf{x}^t))\|_2^2]$$
$$+ \beta_1^2 \mathbb{E}[\|\hat{\mathbf{g}}^t - \mathbb{E}[\hat{\mathbf{g}}^t]\|_2^2]. \tag{16}$$

For the first term, applying Jensen's inequality yields

$$\mathbb{E}[\|(1 - \beta_1)(\mathbf{m}^{t-1} - \nabla f(\mathbf{x}^t) + \beta_1(\mathbb{E}[\hat{\mathbf{g}}^t] - \nabla f(\mathbf{x}^t))\|_2^2]$$
$$\leq (1 - \beta_1)\mathbb{E}[\|\mathbf{m}^{t-1} - \nabla f(\mathbf{x}^{t-1}) - \nabla f(\mathbf{x}^t) + \nabla f(\mathbf{x}^{t-1})\|_2^2] + \beta_1\mathbb{E}[\|\mathbb{E}[\hat{\mathbf{g}}^t] - \nabla f(\mathbf{x}^t)\|_2^2]. \tag{17}$$

By Young's inequality, we have

$$\mathbb{E}[\|\mathbf{m}^{t-1} - \nabla f(\mathbf{x}^{t-1}) - \nabla f(\mathbf{x}^t) + \nabla f(\mathbf{x}^{t-1})\|_2^2] \leq \left(1 + \frac{\delta\beta_1}{2}\right)\mathbb{E}[\|\mathbf{m}^{t-1} - \nabla f(\mathbf{x}^{t-1})\|_2^2]$$
$$+ \left(1 + \frac{2}{\delta\beta_1}\right)\mathbb{E}[\|\nabla f(\mathbf{x}^t) - \nabla f(\mathbf{x}^{t-1})\|_2^2]. \tag{18}$$

For the second term, applying Cauchy's inequality yields

$$\mathbb{E}[\|\hat{\mathbf{g}}^t - \mathbb{E}[\hat{\mathbf{g}}^t]\|_2^2] \leq 3\mathbb{E}\|\hat{\mathbf{g}}^t - \mathbf{g}^t\|_2^2 + 3\mathbb{E}[\|\mathbf{g}^t - \nabla f(\mathbf{x}^t)\|_2^2] + 3\mathbb{E}[\|\nabla f(\mathbf{x}^t) - \mathbb{E}[\hat{\mathbf{g}}^t]\|_2^2]$$
$$\leq 6(1 - \delta)\mathbb{E}[\|\nabla f(\mathbf{x}^t)\|_2^2] + 3(2 - \delta)\sigma^2, \tag{19}$$

where the last inequality uses Assumption 4.3 and 4.4. Applying equation 17equation 18equation 19 to equation 16 and using Assumption 4.2 and 4.4, we obtain

$$\mathbb{E}[\|\mathbf{m}^t - \nabla f(\mathbf{x}^t)\|_2^2] \leq \left(1 - \beta_1\left(1 - \frac{\delta}{2}\right)\right)\mathbb{E}[\|\mathbf{m}^{t-1} - \nabla f(\mathbf{x}^{t-1})\|_2^2] + \frac{2L^2}{\delta\beta_1}\mathbb{E}[\|\mathbf{x}^t - \mathbf{x}^{t-1}\|_2^2]$$
$$+ (\beta_1 + 6\beta_1^2)(1 - \delta)\mathbb{E}[\|\nabla f(\mathbf{x}^t)\|_2^2] + 3(2 - \delta)\beta_1^2\sigma^2. \tag{20}$$

Summing equation 20 for $t = 1, 2, \cdots, T$ yields equation 15. □

Now we are ready to prove Theorem 4.5. We first restate the theorem below in Theorem A.2.

**Theorem A.2.** *Under Assumptions 4.1-4.4, if $\beta_1 \in (0, \delta/(24 - 12\delta))$ and $\eta \leq \min\{1/2L, \sqrt{(\delta\beta_1^2)/(8L^2)}\}$, CE-LoRA with momentum SGD converges as*

$$\frac{1}{T+1}\sum_{t=0}^{T} \mathbb{E}[\|\nabla f(\mathbf{x}^t)\|_2^2] \leq \frac{8[f(\mathbf{x}^0) - \inf_{\mathbf{x}} f(\mathbf{x})]}{\delta\eta(T+1)} + \frac{8\|\mathbf{m}^0 - \nabla f(\mathbf{x}^0)\|_2^2}{\delta\beta_1(T+1)} + \frac{24\beta_1\sigma^2}{\delta}. \tag{21}$$

*Proof.* By Assumption 4.2, we have

$$
\begin{aligned}
f(\mathbf{x}^{t+1}) - f(\mathbf{x}^t) \leq & \langle \nabla f(\mathbf{x}^t), \mathbf{x}^{t+1} - \mathbf{x}^t \rangle + \frac{L}{2}\|\mathbf{x}^{t+1} - \mathbf{x}^t\|_2^2 \\
= & \left\langle \frac{\mathbf{m}^t}{2}, \mathbf{x}^{t+1} - \mathbf{x}^t \right\rangle + \left\langle \nabla f(\mathbf{x}^t) - \frac{\mathbf{m}^t}{2}, \mathbf{x}^{t+1} - \mathbf{x}^t \right\rangle + \frac{L}{2}\|\mathbf{x}^{t+1} - \mathbf{x}^t\|_2^2 \\
= & -\left( \frac{1}{2\eta} - \frac{L}{2} \right)\|\mathbf{x}^{t+1} - \mathbf{x}^t\|_2^2 + \frac{\eta}{2}\|\nabla f(\mathbf{x}^t) - \mathbf{m}^t\|_2^2 - \frac{\eta}{2}\|\nabla f(\mathbf{x}^t)\|_2^2.
\end{aligned} \tag{22}
$$

Taking expectation and summing equation 22 for $t = 0, 1, \cdots, T$ yields

$$
\begin{aligned}
\inf_{\mathbf{x}} f(\mathbf{x}) - f(\mathbf{x}^0) \leq & \frac{\eta}{2} \sum_{t=0}^{T} \mathbb{E}[\|\nabla f(\mathbf{x}^t) - \mathbf{m}^t\|_2^2] - \left( \frac{1}{2\eta} - \frac{L}{2} \right) \sum_{t=0}^{T} \mathbb{E}[\|\mathbf{x}^{t+1} - \mathbf{x}^t\|_2^2] \\
& - \frac{\eta}{2} \sum_{t=0}^{T} \mathbb{E}[\|\nabla f(\mathbf{x}^t)\|_2^2].
\end{aligned} \tag{23}
$$

Applying Lemma A.1 to equation 23 and noting that $\beta_1 \in (0, \delta/(24 - 12\delta))$ implies $(1 - \delta/2)(1 + 6\beta_1) \leq 1 - \delta/4$, we obtain

$$
\begin{aligned}
\frac{1}{T+1} \sum_{t=0}^{T} \mathbb{E}[\|\nabla f(\mathbf{x}^t)\|_2^2] \leq & \frac{8[f(\mathbf{x}^0) - \inf_{\mathbf{x}} f(\mathbf{x})]}{\delta\eta(T+1)} + \frac{8\|\mathbf{m}^0 - \nabla f(\mathbf{x}^0)\|_2^2}{\delta\beta_1(T+1)} + \frac{24\beta_1\sigma^2}{\delta} \\
& - \frac{8}{\delta\eta} \left( \frac{1}{2\eta} - \frac{L}{2} - \frac{2\eta L^2}{\delta\beta_1^2} \right) \sum_{t=0}^{T} \|\mathbf{x}^{t+1} - \mathbf{x}^t\|_2^2.
\end{aligned} \tag{24}
$$

Since $\eta \leq \min\{1/2L, \sqrt{(\delta\beta_1^2)/(8L^2)}\}$ implies $1/(4\eta) \geq L/2$ and $1/(4\eta) \geq (2\eta L^2)/(\delta\beta_1^2)$, equation 21 is a direct result of equation 24. $\qquad \square$

