# OpenReview forum: "CE-LoRA: Computation-Efficient LoRA Fine-Tuning for Large Language Models"
_TMLR — Withdrawn by Authors_

### Review · Reviewer_Tzgd · 2026-02-17

**Summary Of Contributions:**

**Contributions**
To reduce the computational overhead of LoRA, this paper proposed computation-efficient (CE)-LoRA, which leverages approximated matrix multiplication (AMM) to selectively calculate the matrix product, and a double-LoRA technique to mitigate the approximation error induced by AMM. Theoretical analysis demonstrates that CE-LoRA achieves the same convergence rate as vanilla LoRA. Numerical evaluation corroborates that CE-LoRA offers up to 36.3% runtime speedup.

**Strengths**
1. The paper is clearly written, and both the methodology and analysis are easy to follow.
2. Using AMM coupled with double-LoRA to approximate the gradient calculation is novel.
3. Rigorous theoretical guarantees are provided to showcase the convergence rate of CE-LoRA.

**Weaknesses**
1. Assumption 4 is overly strong from an optimization viewpoint, as the gradient error is associated with the choice of $p$ and $r_0$. It would be beneficial if some theoretical bounds can be provided.
2. AMM updates the importance scores every $\tau$ iterations. While $W_s$ is frozen during the fine-tuning, the gradient $G_y$ may vary significantly across mini-batches. Additional empirical justification is needed to explain why infrequent updates of importance scores remain effective in this setting.
3. While CE-LoRA can save **up to** 36.3% runtime, it requires ~25% more iterations to reach a comparable loss to LoRA, which partially offsets the claimed efficiency gains. Moreover, Tables 2 and 3 show a noticeable performance drop. I would suggest including additional experiments comparing the performances under the same time budget.
4. The reported 36.3% runtime reduction is achieved when the matrix dimension is 8192 and the sparsity level is 1/16. The former is less common than 2048 and 4096 in many widely used models (e.g., LLaMA2-7B adopted in the test uses a hidden size of 4096). The latter could introduce a large error. It is suggested to evaluate more commonly used matrix sizes and moderate sparsity levels.
5. Section 5.3 suggests that part of the efficiency gain of CE-LoRA comes from specialized CUDA kernels tailored specifically to the proposed method. This raises concerns about the fairness of the comparison with vanilla LoRA. It is unclear whether similarly optimized kernels were implemented for LoRA. The authors should clarify the implementation details and provide ablation studies that separate the algorithmic improvements from kernel-level engineering optimizations.

**Additional Comments:**

1. Some related works on accelerating the convergence of LoRA are missing, including LoRA-Pro [1], LoRA-RITE [2], and RefLoRA [3].

2. Minor points: (i) In the last sentence of the second paragraph of Section 1, "computation-efficient **fine-tune** algorithm" should be revised to "computation-efficient **fine-tuning** algorithm." ii) Most citations are in the wrong (textual instead of parenthetical) format. iii) Some notations are inconsistent/incorrect. For example, in the sentence above Eq. (10), the double-LoRA rank should be $r_0$ instead of $r$, and $\Sigma$ should also be truncated accordingly.

[1] Z. Wang, J. Liang, R. He, Z. Wang, and T. Tan, "LoRA-Pro: Are Low-Rank Adapters Properly Optimized?", in ICLR 2025.
[2] J. Yen, S. Si, Z. Meng, F. Yu, S. Duvvuri, I. Dhillon, C. Hsieh, and S. Kumar, "LoRA Done RITE: Robust Invariant Transformation Equilibration for LoRA Optimization," in ICLR 2025.
[3] Y. Zhang, B. Li, and G. Giannakis, "RefLoRA: Refactored Low-Rank Adaptation for Efficient Fine-Tuning of Large Models," in NeurIPS 2026.

**Audience:**

Yes

**Audience Explanation:**

People working on parameter-efficient fine-tuning including LoRA will be interested in knowing this paper.

**Claims And Evidence:**

No

**Claims Explanation:**

1. Contribution 3 is an over-statement. The claimed 3.39× acceleration applies only to backpropagation computation, whereas the overall end-to-end speedup is only up to 36.3%, which is acquired under a dimension of 8192 and sparsity level of 1/16. Moreover, this speedup depends on hyperparameter choices that trade accuracy for efficiency.
2. For double-LoRA, I don't fully agree with the claim "The first part $W_s$ is believed to have better structured sparsity and is more compatible to AMM". Being low-rank does not necessary mean certain outer-product $p_i q_i^\top$ are less important or more amenable to structured sparsification. However, I do agree that $W_s^\top G_y$ contributes less to $G_{x,2}$ than $A_0^\top B_0^\top G_y$.

**Requested Changes:**

See weaknesses, claims, and additional comments.

---

### Review · Reviewer_5kUq · 2026-02-27

**Summary Of Contributions:**

This paper focuses on reducing the computational complexity of LoRA. Specifically, LoRA requires full-parameter multiplication when computing the output in the forward pass and the input gradient in the backward pass. To reduce the computational complexity, the paper proposes using approximated matrix multiplication (AMM) combined with Low-rank + sparse decomposition to the frozen matrix.

Strength:
1. The proposed model structure can reduce the runtime of the foundation models during backward propagation.
2. The paper provides the convergence analysis of the proposed algorithm by adopting the widely used error-contraction assumption in gradient compression.
3. The paper provides empirical justification for the efficiency of the proposed algorithm.

Weakness:
1. The proposed sparse + low-rank structure of the weight has been used in SLTrain (Han et al., 2024a). The paper failed to provide a justification for its novelty compared with existing algorithms.
2. Based on the experiment results in Figure 5 and Table 2, CE-LoRA consistently underperforms LoRA in all tasks. It takes longer for CE-LoRA to reach the same loss/accuracy level. Therefore, reducing the per-iteration computation complexity might not benefit the overall training.
3. The proof for the algorithm is widely used in analyzing compressed gradients for communication-efficient optimization. The paper failed to provide a theoretical justification of Assumption 4.4.

4. In addition, the proposed model only reduces the computation efficiency in backward propagation, while forward propagation still requires dense matrix multiplication of the same size, indicating that the per-iteration runtime reduction is no more than 50%.

**Audience:**

Yes

**Audience Explanation:**

CE-LoRA is a timely contribution to the field of Parameter-Efficient Fine-Tuning (PEFT). While most recent work (like QLoRA or DoRA) focuses on memory or quality, CE-LoRA tackles the computational bottleneck of the backward pass without breaking the core benefits of LoRA.

**Claims And Evidence:**

No

**Claims Explanation:**

The numerical results are insufficient to support the claim that the proposed method can reduce computational complexity. As stated in the weakness above, the results show that the reduction in each iteration comes at the cost of slower convergence. The paper failed to provide evidence that the model achieves the same level of model performance with less computation.

**Requested Changes:**

Figure 6: The figure contradicts the statement. The statement says CE-LoRA speeds up the backward pass, but the figures show that LoRA and CE-LoRA have the same BW time, while CE-LoRA uses less FW time.

---

### Review · Reviewer_C1bg · 2026-03-07

**Summary Of Contributions:**

The paper propose CE-LoRA, which reduces the computational cost of LoRA by randomly drop a fraction of matrix multiplications during forward and backward. Experiments in finetuning task show that CE-LoRA requires less training time while maintaining reasonable downstream task performance compared to standard LoRA.

**Strengths:**
* **Lower computational FLOPs.** CE-LoRA reduces computational FLOPs by randomly dropping a fraction of matrix multiplications during backward. The advantage of lower FLOPs leads to faster training process.
* **Reasonable performance.** While the convergence is slower than standard LoRA due to approximate backpropagation, the downstream performance is close to LoRA, making it a candidate for finetuning when computational resource is a bottleneck.

**Weaknesses:**
* **Improved memory cost.** CE-LoRA requires additional storage of low-rank matrices that are the principle components of the base model weight. The improved memory cost contradicts with the motivation of using LoRA to reduce memory profile.
* **Lack of in-time convergence comparison.** Since the main motivation of CE-LoRA is to reduce the training FLOPs of LoRA, it is essential to provide the in-time loss convergence comparison between CE-LoRA and LoRA.
* **Impractical experiments setup.** When using LoRA, it is essential to adopt mixed-precision training (i.e. use float32 for low-rank matrices) for fast convergence, rather than using pure16 training. Thus, It is suggested to run all the experiments in mixed-precision training to match the usage in practice.

**Questions:**
* In Figure 6, the backward time of LoRA and CE-LoRA are almost identical; the result misalign with the FLOPs reduction of CE-LoRA. Moreover, the backward cost should be heavier than forward cost, yet the figure shows that its time cost is less, which is a bit confusing to me.
* The idea of dropping a fraction of matrix multiplication is not tailored to LoRA training, but general LLM training. Is there any special problem structure that makes the technique more suitable for LoRA training?

**Audience:**

Yes

**Audience Explanation:**

Reducing the computational cost of post-training is of general interest of the community. Approximate matrix multiplication mentioned in the paper can be a potential research direction, especially when it utilizes problem structure when determining the approximation strategy.

**Broader Impact Concerns:**

Not applicable.

**Claims And Evidence:**

No

**Claims Explanation:**

Please see "Summary of contributions".

**Requested Changes:**

I suggest the authors to take the following changes to their manuscript for strengthening the work:
1. Include in-time convergence comparison of CE-LoRA and LoRA. Since CE-LoRA induces higher memory cost, the method is valuable only when its in-time convergence is better.
2. Adopt mixed-precision training when benchmarking the algorithms, which matches the usage of LoRA in practice.
3. Experiment with larger models. The approximate matrix multiplication induces higher errors when the weight matrices have higher dimension. Hence, to justify the effectiveness of the approach, it is critical to verify it in larger-scale models.

---

### Note · Authors · 2026-03-11

**Comment:**

We have decided to withdraw this submission for further improvement. We sincerely thank the reviewers for their time, effort, and valuable feedback.

**Withdrawal Confirmation:**

I have read and agree with the venue's withdrawal policy on behalf of myself and my co-authors.